# Impact of Long Working Hours on Mental Health Status in Japan: Evidence from a National Representative Survey

**DOI:** 10.3390/ijerph21070842

**Published:** 2024-06-27

**Authors:** Xinxin Ma, Atushi Kawakami, Tomohiko Inui

**Affiliations:** 1Faculty of Economics, Hosei University, 4342 Machita-shi Aiharamachi, Tokyo 194-0298, Japan; 2Faculty of Economics, Toyo University, Tokyo 112-8606, Japan; kawakami@toyo.jp; 3Faculty of International Social Sciences, Gakushuin University, Tokyo 171-8588, Japan; tomohiko.inui@gakushuin.ac.jp

**Keywords:** mental health, long working hours, regular worker, non-regular worker, Japan

## Abstract

Using the 2010–2019 Comprehensive Survey of Living Conditions (CSLC) conducted in Japan, we examined the impact of long working hours on mental health in Japan while addressing the endogeneity issue arising from non-random selection bias. We assessed the variations in the effects of long working hours on mental health across different groups. The results show that first, individuals working longer hours (55 h or more per week) exhibited a higher likelihood of developing mental illness than those working regular hours or fewer hours. Second, the negative effect of long working hours on mental health is more pronounced among non-regular workers than among regular workers. Third, the effect of long working hours on mental health varies among different demographic groups, with a greater impact observed among women, managers, non-regular workers, employees in small- or large-sized firms, and those in smaller cities compared to their counterparts. Thus, to enhance worker productivity, the Japanese government should address the issue of long working hours to improve employees’ mental well-being. Initiatives aimed at promoting work–life balance, family-friendly policies, and measures to ameliorate working conditions are expected to help mitigate the challenges associated with long working hours and mental health issues, especially among non-regular workers.

## 1. Introduction

Mental illness is often referred to as a mental disorder, and long working hours are two significant issues affecting employees worldwide [1,2,3]. In 2017, approximately 792 million individuals worldwide were reported to be living with mental illness, accounting for 10.7% of the global population, slightly exceeding 1 in 10 people [2]. Given the high medical care expenses associated with mental illness [2,3], and considering that mental illness can reduce labor productivity, thereby negatively affecting human capital accumulation in most countries, exploring the determinants of mental illness has become a critical issue in the field of public health.

Several empirical studies have shed light on potential determinants of mental illness, including individual attributes such as education, age, and sex [4,5,6,7]. Additionally, social capital has been identified as a factor that influences the risk of developing mental illness [8,9], along with life events such as marriage and fertility [10,11,12]. Understanding these factors can contribute significantly to addressing mental illness challenges and implementing effective public health interventions.

Furthermore, work–life conflict has been identified as a factor that may increase the risk of mental illness [13,14]. Empirical research has consistently highlighted long working hours as a significant risk factor among the various work environment factors that can adversely affect workers’ mental health. Numerous studies have found a strong association between long working hours and a higher risk of developing mental disorders in both developing and developed countries [4,5,6,7,15,16,17,18]. However, the evidence from Japan is scarce.

In Japan, the prevalence of workplace stress among workers is steadily increasing, which has implications for mental health. According to the Survey on the State of Employees’ Health conducted by the Ministry of Health, Labour and Welfare (MHLW), the percentage of workers experiencing strong anxiety, worry, or stress in their work or working life increased from 55.0% in 1987 to 61.5% in 2002. This rise in workplace stress is also evident in the alarming increase in the number of patients with mental disorders and suicides nationwide. The total number of patients with mental disorders in Japan increased from 4.33 million in 2013 to 17.21 million in 2022. The total number of suicides in Japan surged from 27,282 in 2013 to 21,881 in 2022, even as the number of employed persons committing suicide decreased from 9401 in 2013 to 7862 in 2016 and increased to 8576 in 2022. The 2008 White Paper on Suicide Countermeasures, published by the Cabinet Office, revealed that “health problems” accounted for the largest proportion of causes and motives for suicide (63.3%). Long working hours in Japan are a critical factor contributing to these issues. In 2004, the MHLW published a report on the “Study Group on Overwork and Mental Health Countermeasures”, which selected employees based on the fact that they had worked long overtime hours. It recommends creating a mechanism to check employee health. Recently, the Japanese government promoted the reduction in long working hours and improved mental health in the workplace. However, when compared internationally, Japan, along with the United States, had the longest average annual working hours of 1800 h or more among developed countries during the 2000s. Therefore, investigating the association between long working hours and mental illnesses, particularly in Japan, is of the utmost importance.

This study focused on this issue in Japan. Some studies have investigated this issue from the perspectives of occupational health and epidemiology. Fujino et al. [19] conducted a systematic literature review of 17 papers that addressed this issue. Recent survey data have been utilized by other researchers [20,21,22,23,24], who have also examined the relationship between long working hours and the likelihood of developing mental disorders, such as depression and stress, in Japan. However, many of these studies, from occupational health or medical perspectives, have not adequately considered endogeneity. However, few empirical studies have explored this issue from an economic perspective. Three of these studies are particularly relevant to this study. Two studies investigated the association between long working hours and mental health, using longitudinal survey data and employing fixed-effects (FE) models to address individual heterogeneity [25,26]. Okamoto used longitudinal survey data and instrumental variable (IV) methods to discover that long working hours may lead to reduced sleep hours and an increased probability of becoming obese [27]. However, these three studies did not adequately account for non-random sample selection issues between the long and non-long working-hour groups. This issue is related to endogeneity and may introduce a measurement bias into the empirical results. Additionally, although there are large disparities in work conditions (e.g., earned wages, work hours, and social insurance) in Japan between regular and non-regular workers (e.g., part-time, temporary, dispatched, contract, and entrusted workers) [28,29,30] and it has been noted that the working hours of non-regular workers are also long in Japan [31], previous studies have not explored the differences in the effects of long working hours on mental illness between regular and non-regular workers. This study addresses these gaps in the literature.

Using national survey data from 2010, 2013, 2106, and 2019 and the propensity score matching (PSM) method, this study investigates the association between long working hours and mental illness in Japan, considering the endogeneity issue arising from non-random selection bias, and compares the effects of long working hours between regular and non-regular workers.

This study makes three notable contributions to the related literature. First, as mentioned above, the gaps in work conditions between regular and non-regular workers in Japan have attracted attention in empirical studies. However, to the best of our knowledge, there is no evidence of differences in the effects of long working hours on mental illness between regular and non-regular workers. In Japan, the working hours of some non-regular workers (especially dispatched, contracted, or entrusted workers) are similar to those of regular workers [31]. Japan can be used as a case study to address this issue. This study is the first to compare the effects of long working hours on mental health status according to employment type in Japan. 

Second, we developed a methodology to investigate the relationship between work hours and mental health status. Unlike most studies that have overlooked the endogeneity issue, we aim to address this limitation. While previous studies [25,26,27] have attempted to tackle this problem using the fixed-effects (FE) model or instrumental variable (IV) method to investigate the causal relationship between long working hours and mental health status, they did not sufficiently account for the endogeneity issue arising from non-random selection bias. Our study employed the propensity score matching (PSM) method to address this concern. To the best of our knowledge, this study is the first to explore the relationship between long working hours and mental health status in Japan based on the PSM method, thereby providing enriching evidence on this issue. Additionally, unlike previous studies that focused on a single definition of “long working hours”, we introduced various definitions (e.g., different cut-off values) to perform robustness checks. We compared the effects of working hours on mental health among different weekly working hour groups (40-, 45-, 50-, 55-, and 60-h groups).

Third, we extended our investigation by comparing the impact of long working hours on mental health among different groups (by gender, occupation, employment status, firm size, and region), offering valuable insights into potential variations in this relationship.

## 2. Literature Review and Hypotheses Development

Regarding the relationship between long working hours and mental illness, two models from the occupational health and personal economics perspectives have been proposed to suggest that long working hours negatively affect mental health status. First, the job demand–control model [32] posits that involuntary long working hours may lead to mental illness because of an imbalance between work responsibility (the reality of long working hours) and authority (wherein employees lack the authority to determine their working hours). Second, according to the effort–reward imbalance model [33], when the efforts involved in involuntary long working hours are not adequately rewarded (e.g., through unpaid overtime or a low overtime premium), the probability of developing mental illness may be higher among those working long hours.

From the perspective of labor and family economics theory based on the individual/household utility model, individuals aim to maximize their utility while considering income and time constraints. An individual’s time allocation can be divided into three parts: market work, housework, and leisure [34,35]. This division implies a trade-off between working hours and hours devoted to housework and leisure activities. Consequently, long working hours may reduce the time available for housework and leisure, leading to work–life conflicts. This conflict may be particularly pronounced for female workers during their motherhood, as they face additional responsibilities and time constraints. This reduction in housework/leisure time and the resulting work–life conflict may ultimately lead to a decline in an individual’s utility and increase the risk of developing mental illness [36,37].

Numerous studies have found that long working hours are a primary contributor to mental illness among working-age populations, in addition to demographic factors [4,5,6,7,15,16,17,18], family-related factors [10,11,12], and social factors such as social participation and social capital [8,9].

Some studies have reported that long working hours negatively affect mental health [4,5,6,7,15,16,17,18]. In Japan, there is a series of studies focusing on this issue from occupational health, epidemiology [19,20,21,22,23,24], and economic perspectives [25,26,27]. Their findings indicated that long working hours may increase the likelihood of developing mental disorders, such as depression and stress in Japan. A few empirical studies have addressed this issue from an economic perspective, three of which are closely related to this study. Ma [25] utilized survey data from the 2004–2008 Keio Household Panel Survey (KHPS 2004–2008) and employed a fixed-effects model with lagged variables to address endogeneity concerns. The empirical analysis investigated the impact of long working hours on the mental health status of workers in Japan and found that long working hours increased the probability of developing mental disorders. Furthermore, the study found that the impact of long working hours on mental health status varies across different groups; it is more significant for workers in the private sector, workers in large firms, low-education workers, low-income workers, female workers, and male non-managerial occupational workers than for their counterparts. Kuroda and Yamamoto [26] used four-wave longitudinal data from the Survey of Companies and Employees on Human Capital Development and Work-Life Balance, which has been conducted annually since 2012 by the Research Institute of Economy, Trade, and Industry. Their investigation focused on how the number of work hours, job characteristics, and workplace circumstances affected workers’ mental health. The study found that long working hours significantly contributed to deterioration in respondents’ mental health status, even after controlling for individual fixed effects and other characteristics. Furthermore, the relationship between work hours and mental health status was not linear, as working more than 50 h per week notably erodes workers’ mental health status. Using longitudinal survey data from the Japan Household Panel Survey/Keio Household Panel Survey of 2004–2017 and the instrumental variable (IV) method, Okamoto [27] analyzed the effects of work hours on body mass index, smoking, and sleeping hours in Japan. This study found that longer working hours led to reduced sleep duration and increased the probability of obesity among workers.

According to personnel and labor economics theories and the empirical studies above, Hypothesis 1 (H1) is formulated as follows:

**Hypothesis** **1.** 
*Individuals working longer hours exhibit a higher likelihood of developing mental illness compared to those working regular or fewer hours.*


Several studies have demonstrated the persistence of the wage gap between regular and non-regular workers in Japan [28,29,30]. Some studies find that the wage gap between the two groups is due to disparities in endowment factors (e.g., human capital) and differences in the return to endowment (e.g., wage return on education), which are related to discrimination against non-regular workers in the workplace [28,30]. According to the effort–reward imbalance model [33], when non-regular workers work similar hours but receive lower wages than their counterparts (regular workers), the negative effect of long working hours on mental health may be more pronounced among non-regular workers. Consequently, Hypothesis 2 (H2) was assumed to be

**Hypothesis** **2.** 
*The negative effect of long working hours on mental health status is more pronounced among non-regular workers than among regular worker groups.*


In the following section, we conduct an empirical analysis to clarify these two hypotheses.

## 3. Empirical Study Methodology

### 3.1. Model

As a benchmark, we employ the ordinary least squares (OLS) method to estimate the association between long working hours and mental illness, as expressed in Equation (1). The ordinary least squares (OLS) method is a type of linear least squares method used to estimate unknown parameters in a linear regression model by minimizing the sum of the squares of the differences between the observed dependent variable in the dataset and the predicted values from the linear function of the independent variables.
(1)$$MI_{i}=a+\beta_{LWH}{WH}_{i}+\beta_{NR}{NR}_{i}+\beta_{LNR}{\mathrm{WH}\times NR}_{i}+\sum_{n}\delta_{n}X_{ni}+\varepsilon_{i}$$
where MI denotes the risk of becoming mental illness; i and n denote the individual and number of covariates, respectively; LWH represents a set of indicators of work hours (e.g., 40, 45, 50, 55, 60 h); NR represents the non-regular workers; WH×NR represents the interaction term of WH and NR; X denotes the covariates; β and δ are the coefficients, respectively, a is a constant term; and ε is an error term.

The groups that worked long hours and those with regular or shorter work hours may not have been randomly selected. Some unobservable variables could influence the probability of long working hours. To address this endogeneity problem, we employed the PSM method, a statistical matching technique that estimates the effect of a treatment (in this case, working long hours). PSM aims to reduce the bias stemming from confounding variables that may affect the treatment effect estimate when comparing outcomes between units that received treatment and those that did not [38]. The propensity score matching method makes the observed data more akin to randomized experimental data through matching and resampling, thereby minimizing selectivity bias and counterfactual states in the sample composition.

Using propensity score matching, we calculated the average treatment effect on the treated group, the average treatment effect on the untreated group, and the overall average treatment effect as follows. First, we selected appropriate control variables for resampling in propensity score matching. Second, we ran a probit regression to estimate the propensity score. Third, we matched propensity scores based on the selected covariates. Finally, we calculated the average treatment effect on the treated (ATT), average treatment effect on the untreated (ATU), and the overall average treatment effect (ATE) based on the matched samples. In this study, we report the results of the ATT.
(2)$$ATT=E({MI}_{it}\left(1\right)-{MI}_{it}\left(0\right)|LWH=1)$$
(3)$$ATU=E\left({MI}_{it}\left(1\right)-{MI}_{it}\left(0\right)|LWH=0\right)$$
(4)$$ATE=E\left(C_{it}\left(1\right)-C_{it}\left(0\right)\right)$$
where ${MI}_{it}\left(1\right)$ expresses the risk of mental illness when an individual works long hours, and ${MI}_{it}\left(0\right)$ represents the risk of mental illness when the same individual does not work long hours. We also used PSM to compare the effect of long working hours on mental health status among different groups using subsamples.

### 3.2. Date and Variable Setting

This study used repeated cross-sectional data from the Comprehensive Survey of Living Conditions (CSLC), a nationally representative survey conducted by MHLW in Japan. The CSLC surveys households and their members throughout the country. We used the latest four waves of the CSLC, from 2010, 2013, 2016, and 2019, which provide rich information on mental health status, individual and household characteristics, and work status. Because the questions in the household questionnaire were nearly identical across these four waves, we pooled the data from the CSLC for these years in our analysis.

To collect household and health records, all households and their members within the 5530 districts were stratified and randomly sampled from Postal Codes 1 and 8 of the National Census. For nursing care records, individuals requiring nursing care and support under the Long-term Care Insurance Act within 2500 districts were stratified and randomly sampled from 5530 districts. Additionally, income and savings certificates were stratified and randomly sampled from Postal Code 1 in the 5530 districts mentioned above. A survey was conducted on all households and their members within a 2000-unit district. The CSLC is a representative national survey on health status, income, and nursing care in Japan.

The dependent variable in the risk of mental illness was the mental health score, which was calculated based on the following six questions:(a)Do you feel nervous?(b)Do you feel hopeless?(c)Do you feel restless?(d)Do you feel depressed and like nothing could clear your mind?(e)Did you experience difficulty in doing anything?(f)Do you feel worthlessness?

We assigned scores to each item based on the following response options: 5 = every time; 4 = often; 3 = sometimes; 2 = not often; and 1 = no. Higher values indicate a greater probability of experiencing mental illness. By summing the scores from the questionnaire items, we calculated the total mental illness score based on the respondents’ answers, which ranged from 5 to 25.

The key independent variable for mental health function was working hours. We created a set of weekly work hours (total number of work hours, including main and side jobs) dummy variables, as follows:(a)40 h (1 = 40 h, 0 = otherwise)(b)45 h (1 = 45 h, 0 = otherwise)(c)50 h (1 = 50 h, 0 = otherwise)(d)55 h (1 = 55 h, 0 = otherwise)(e)60 h (1 = 60 h, 0 = otherwise)

The covariate variables were utilized in mental health function based on the OLS method. To account for the differences in work conditions between the regular and non-regular work groups, we created a regular worker dummy variable (1 = regular worker, 0 = non-regular worker). We defined non-regular workers based on the CSLS questionnaire items of CSLS including part-time, temporary worker, dispatched, contract, and entrusted workers. Additionally, we incorporated an interaction term between working hours and the regular worker dummy variable to estimate the difference in the effect of long working hours on mental health status between the two groups.

We utilized the following variables in the probability function for matching:(1)A female dummy variable (1= female, 0 = male) was used to control for the gender gap in work hours, as numerous studies have reported that work hours differ by gender, with men generally working longer hours than women.(2)To account for potential differences in work hours among age groups (younger, middle-aged, and older), we included age as a variable in our analysis.(3)The co-residence relationship dummy variable was used to control for the influence of family members on mental health status.(4)Occupational dummy variables were employed in the analysis to categorize workers into nine types of occupations: (a) manager; (b) professional job; (c) clerks; (d) sales job; (e) service job; (f) security job; (g) agriculture, forestry, and fishery job; (h) elementary job; and (i) other occupations.(5)To account for the influence of firm size on mental health status, we included eight types of firm size dummy variables: (a) 1–29 employees, (c) 30–99 employees, (d) 100–299 employees, (e) 300–499 employees, (f) 500–999 employees, (g) 1000–4999 employees, (h) 5000 or more employees, and (i) government offices.(6)We consider the impact of non-earned income on labor supply and household income.(7)The spouse’s employment status was categorized using seven types of dummy variables: (a) regular worker, (b) part-time worker, (c) temporary worker, (d) dispatched worker, (e) contract worker, (f) entrusted worker, and (g) other employment status excepting the above types.(8)To control the influence of childcare on working hours, we constructed a variable representing the number of children.(9)We created five types of dummy variables representing regions based on the population in cities to control for the influence of city size on mental health status: (a) city with a population of less than 50 thousand; (b) city with a population of 50–149 thousand; (c) city with a population of 150 thousand; (d) large city; and (e) countryside.

## 4. Descriptive Statistics Results

Descriptive statistics for the variables are presented in Table 1. We compared the mean value gap for each variable between the group with weekly work hours of 55 or more (long working hours [LWH]) and the group with weekly work hours of less than 55 (non-long working hour [non-LWH]).

First, the mental health score of the LWH group (3.536) was higher than that of the non-LWH group (3.204), suggesting a higher probability of mental illness among the LWH group than among the non-LWH group.

Second, individual attributes (gender and age), family structure factors (number of children and household income), work-related factors (occupation and firm size), and region differed between the two groups. For example, the proportion of regular workers in the LWH group (93.0%) was higher than that in the non-LWH group (58.6%), and the proportion of female workers in the LWH group (12.1%) was lower than that in the non-LWH group (48.9%). The proportion of managers in the LWH group (11.4%) is higher than that in the non-LWH group (6.5%). These differences suggest that these factors may potentially influence the probability of being a worker with long working hours, which can lead to an endogeneity issue due to the non-random selection between the long working hours group and the regular working hours group. Therefore, this issue must be addressed in future studies.

Table 2 presents the average work hours across the heterogeneous groups. The results revealed that weekly working hours differed among the groups. It is observed that weekly working hours are longer for male workers, middle-aged workers (aged 30–49), well-educated workers, managers, and regular workers than for their counterparts (female workers, younger and older generations, low- and middle-level educated workers, non-managerial workers, and non-regular workers). It is assumed that the effects of long working hours differ among these groups. Hence, in this study, we examined the impact of long working hours on mental health in these heterogeneous groups.

The raw relationship between long working hours and mental illness is shown in Figure 1. First, the mental illness score among non-regular workers was higher than that among regular workers, indicating a higher probability of mental health disorders among non-regular workers compared to their counterparts. Second, the relationship between working hours and mental illness varied between regular and non-regular workers. For regular workers, the mental illness score tended to increase with an increase in working hours in the range of 0–10 h, while it tended to decrease with an increase in working hours in the range of 10–55 h. However, the mental illness score tended to increase with the increase in working hours when working hours exceeded 55 h.

Although descriptive statistics show differences in individual attributes between workers working long and non-long working hours, variations in work hours across heterogeneous groups, and differences in the relationship between work hours and mental illness between regular and non-regular workers, these results do not account for other factors that may influence mental health status. In the following section, we investigate the association between long working hours and mental illness while considering other factors to obtain a more comprehensive understanding of this relationship.

## 5. Econometric Analysis Results

### 5.1. Results Based on the OLS Method

Table 3 presents the results obtained using the OLS method. To compare the impact of long working hours on mental illness between regular and non-regular workers, we included an interaction term between working hours and non-regular worker dummy variables in the analysis. Additionally, considering that the impact of working hours on mental health status may vary depending on the duration of working hours, we used a set of working hour dummy variables (40, 45, 50, 55, and 60 h), with the reference group being 45 h.

We distinguished five models using different indicators of work hours, the interaction term of work hours, and the regular worker dummy variable, as follows: (1)Model 1: Using the working hours variable and excluding the interaction term.(2)Model 2: Using the 40-h dummy variable and including the interaction term.(3)Model 3: Using the 50-h dummy variable and including the interaction term.(4)Model 4: Using the 55-h dummy variable and including the interaction term.(5)Model 5: Using the 60-h dummy variable and including the interaction term.

For Models 2–5, we used 45 h as the reference group. The main findings of this study are as follows.

First, regarding the impact of work hours on mental health status, the coefficients of work hours in Models (2)–(4) were all negative and significant at the 1% level, suggesting that when work hours were 40 or more, the probability of experiencing mental illness increased. Comparing the magnitude of the coefficients for the 40-, 50-, 55-, and 60-work-hour dummy variables, the coefficient is greatest for 60 or more hours (1.994) and lowest for 40 or more hours (0.695), indicating that as work hours increase, the probability of experiencing mental illness significantly increases. These results support Hypothesis 1 (H1).

Second, the coefficients of the regular worker dummy variables in Models (2) to (5) are negative and significant at the 1% level. Additionally, the interaction terms of work hours and the regular worker dummy variable are negative and significant at the 1% level in Models (1) to (4). These results indicate that the impact of long working hours on mental health status is greater for non-regular workers than for regular workers when other factors are held constant. These results support Hypothesis 2 (H2). 

### 5.2. Results Based on the PSM Method

Table 4 and Table 5 present the results for the probability of regular workers (Table 4) and non-regular workers (Table 5) working long hours, respectively. Columns 1–4 present the results for (1) 40, (2) 50, (3) 55, and (4) 60 h, respectively. We used 45 h as the cutoff value to construct the control group using PSM. The variables used for matching included sex, age, co-resident relationship, employment status, occupation, firm size, income, number of children, and city size. The results indicate that the probability of individuals working long hours is higher for men, managers, workers in small- or medium-sized firms, non-regular workers, and those residing in large cities in Japan. These results were used to construct the matched samples.

Table 6 summarizes the average treatment effects for the control and treatment groups. Based on the findings in Table 1, Table 2 and Table 3 and Figure 1, we defined the control group as individuals who worked 45 h or less. We conducted three estimations for 50, 55, and 60 h and used two models: Model 1, which excluded covariates, and Model 2, which included covariates.

All results indicate that, compared to the group working 45 h or less per week, the probability of experiencing a mental disorder increases for the group working 55 h or more and the group working 60 h or more. For instance, for the group working 55 h or more (Estimation 2), in Model 1, the gap in mental illness score between the treatment group (55 h or more group) and control group (45 h or less) was 0.353 and significant at the 10% level, indicating that compared to the group working 45 h or less per week, the probability of experiencing a mental disorder increases for the group working 55 h or more per week. In Model 2, the gap was 0.454 and significant at the 10% level. The results also support the finding that working 55 h or more per week increases the probability of developing mental disorders. These results support Hypothesis 1 (H1) again.

### 5.3. Results by Regular and Non-Regular Workers Based on the PSM Method

The results for regular and non-regular workers are presented in Table 7, based on Model 2 of Estimation 2 in Table 6. We reran the estimations using the PSM method for employment groups.

The results indicated that long working hours (LWH) negatively affected the mental health of both regular and non-regular workers, with the effect of LWH being greater among non-regular workers (0.518) than among regular workers (0.457). These findings are consistent with those presented in Figure 1 and Table 3. Thus, H2 was confirmed.

### 5.4. Results by Gender and Work-Related Groups Based on the PSM Method

The results by gender and work-related groups are presented in Table 8 based on Model 2 of Estimation 2 in Table 6. We reran the estimations using PSM for gender (panel [a]), occupation (panel [b]), firm size (panel [c]), and city (panel [d]) groups. The main findings are as follows.

First, long working hours (LWH) negatively affected the mental health status of both female and male workers, with the effect of LWH being greater among female workers (0.718) than male workers (0.430).

Second, LWH negatively affected the mental health status of both managers and non-managers, with the effect of LWH being greater among managers (0.767) than non-managerial workers (0.430).

Third, LWH negatively affects the mental health status of workers in small, middle, and large firms or government offices, with the effect of LWH being especially pronounced among workers in small firms (with 1–99 employees) and large firms (with 300 or more employees) compared to other groups.

Finally, LWH negatively affects the mental health status of workers in small, medium, and large cities, with the effect of LWH being greater among workers in middle- and small-sized cities than among those in large cities.

## 6. Discussion

### 6.1. Novel Findings of This Study

This study investigated the effects of long working hours on mental health status in the Japanese context, considering a non-random sample selection bias among groups with different working hours. The results indicated that long working hours, especially weekly work hours exceeding 55 h, significantly increased the likelihood of developing mental health disorders. These findings align with previous studies in Japan from occupational health and epidemiological perspectives [19,20,21,22,23,24], as well as from an economic perspective [25,26,27], highlighting the persistent negative impact of long working hours on individuals’ well-being. Furthermore, the results are consistent with conclusions drawn from research conducted in other countries [4,5,6,7,15,16,17,18].

We also compared the effects of long working hours on the mental health status of regular and non-regular workers, a comparison not previously considered in the literature. These new findings suggest that the negative effect of long working hours is more pronounced among non-regular workers than among regular workers. These findings can be explained by the effort–reward imbalance model [33], which suggests that when the efforts involved in involuntary long working hours are not adequately rewarded, the likelihood of experiencing mental illness may be higher among those working long hours. In Japan, there exists a wage gap between regular and non-regular workers [28,29,30]. When non-regular workers receive lower wages despite working similar hours to regular workers, the probability of experiencing mental illness among non-regular workers is likely to be higher than that among regular workers.

Additionally, the results regarding the differences in the effects of long working hours among gender, occupation, firm size, and city size groups demonstrated that the negative effects on mental health status were greater for women, managers, workers in small or large firms, and workers in small cities. We consider the reasons as follows: First, according to family economics theory, there exists a gender role division in households known as “the man as the breadwinner and the woman as the homemaker” [34,35]. As women often bear more responsibility for family care (e.g., childcare and elder care), the negative effect of long working hours on mental health status may be greater for women than for men. Second, in many Japanese firms, managers often do not receive overtime allowances, whereas non-managerial workers do. According to the effort–reward imbalance model [33], this disparity may lead to a greater negative effect of long working hours on managers than on non-managers. Third, there is a wage gap based on firm size in Japan [39,40], with workers in small firms typically earning lower wages than those in medium-sized firms. According to the effort–reward imbalance model, this wage disparity may result in a greater negative effect of long working hours for workers in small firms than in medium-sized firms. Although wages are higher in large firms than in medium firms, workers in large firms often face higher work responsibilities and lower work authority. According to the job demand–control model [32], this imbalance may lead to a greater negative effect of long working hours on workers’ mental health status in large firms than in medium firms. Finally, fierce business competition is more prevalent in large cities. Due to the intense competitive environment, the likelihood of achieving work–life balance may be lower for workers in large cities. Additionally, workers who prefer to avoid fierce competition may choose to work in smaller cities. Therefore, the negative effects of long working hours on mental health may be greater for workers in small cities than those in large cities.

These new empirical findings enrich our understanding of the related personnel and labor economic theories on this issue. This study also underscores the importance of distinguishing between different types of employment status (regular and non-regular workers) and heterogeneous groups (e.g., gender, occupation, firm size, and city size) for future research.

### 6.2. Practical Implications

The practical implications based on the empirical study can be considered as follows:

First, our findings suggest that, compared to the group with 45 work hours or less, the group with 55 work hours or more potentially has a higher risk of developing mental disorders. The results support the policy of promoting weekly work hours of less than 45, as enforced by the Japanese government (e.g., MHLW) in the current period. As the results indicate that weekly working hours exceeding 55 h will increase the risk of mental illness, setting the upper limit for long working hours at 55 h in work-hour regulations may be advisable.

Second, the results indicated that the negative effect of long working hours on mental health among non-regular workers was greater than that among regular workers. Most non-regular workers have less job authority than their counterparts (regular workers), and there exists a significant wage gap between regular and non-regular workers [28,29,30,39]. Based on the job demand–control model [32] and the effort–reward imbalance model [33], long working hours may harm the mental health status of non-regular workers to a greater extent than that of regular workers. Japan faces the issue of the working poor due to the substantial wage gap between non-regular and regular groups among non-regular workers with long working hours [31]. A policy aimed at reducing the wage gap between the two groups is expected to mitigate the negative effects of long working hours, especially for non-regular workers. Meanwhile, implementing work–life balance initiatives, family-friendly policies, and measures to improve working conditions are expected to enhance overall mental health status. 

Third, regarding the determinants of becoming long-hour employees, our results (see Table A1) suggest that, in general (as shown in Column 1, which includes the total sample of both men and women), younger workers aged 16–29, middle-aged and older workers aged 40 and over, workers with low household income, widowers, workers with lower levels of education, non-managers, employees in micro-firms with one to four employees, workers with a spouse who is a regular employee, and those working in large cities with populations of 15 million or more are at a higher risk of becoming long-hour employees than their counterparts. These counterparts include younger senior workers aged 30–39 years, married workers, employees with a high household income, workers who graduated from senior high school, managers, employees working in middle- and large-sized firms, workers whose spouses do not work, and those working in middle- and small-sized regions. The results for the female and male groups (Columns 2 and 3) are similar to those observed in the total sample. The results suggest that workers with young children and disadvantaged workers (those without spouses, low-income workers, individuals with lower educational levels, non-managers, and employees in small firms) are more likely to have long work hours. The differences in socioeconomic status contribute to inequality in work hours between disadvantaged and advantaged groups, indicating that the issue of working poverty may be more severe among disadvantaged groups than advantaged groups.

Policies aimed at reducing working hours may lead to lower earned income levels, potentially worsening the life situations of disadvantaged groups. Therefore, it is essential to emphasize policies aimed at improving wages or earned income, such as reducing the wage gap between regular and non-regular workers (for example, enforcing the implementation of an equal pay for equal work policy), alongside the implementation of work-hour regulation policies.

## 7. Conclusions

Using the long-term Comprehensive Survey of Living Conditions (CSLC), a nationally representative survey conducted in Japan from 2003 to 2019, we examined the impact of long working hours on mental health in Japan while addressing the endogeneity issue arising from non-random selection bias. Additionally, we assessed variations in the effect of long working hours on mental health across different groups, especially regular versus non-regular workers, which has not been analyzed in the existing literature.

This study yields three primary conclusions. First, individuals working longer hours (55 h or more weekly) have a higher probability of developing mental illness than those working regular hours or fewer hours. Consequently, H1 is supported. Second, the negative effect of long working hours on mental health was greater for regular workers than for non-regular workers. These results confirm H2. These conclusions were confirmed by the results obtained using PSM. Third, the effect of long working hours on mental health differs among other various groups by sex and workplace. For instance, it is greater for women, managers, workers in small or large firms, and workers in small cities than for their counterparts.

It should be noted that this study has the following limitations. First, long working hours may have effects on physical health (e.g., chronic diseases) beyond mental health, which should be explored in future research. Second, although we controlled for covariate variables as thoroughly as possible based on the CSLC, some factors (such as the spouse’s attributes and work situation, housing, personal relationships, and work conditions in the workplace) may potentially affect an individual’s mental health. Because we could not obtain this information from the CSLC dataset used in this study, exploring the empirical relationship considering these factors is a future research avenue. Finally, as this study utilized repeated cross-sectional survey data, we were unable to examine the dynamic changes in the effect of long working hours on mental health or address the individual heterogeneity problem. Conducting a study based on panel data would be valuable to address this issue in the future.

## Figures and Tables

**Figure 1 ijerph-21-00842-f001:**
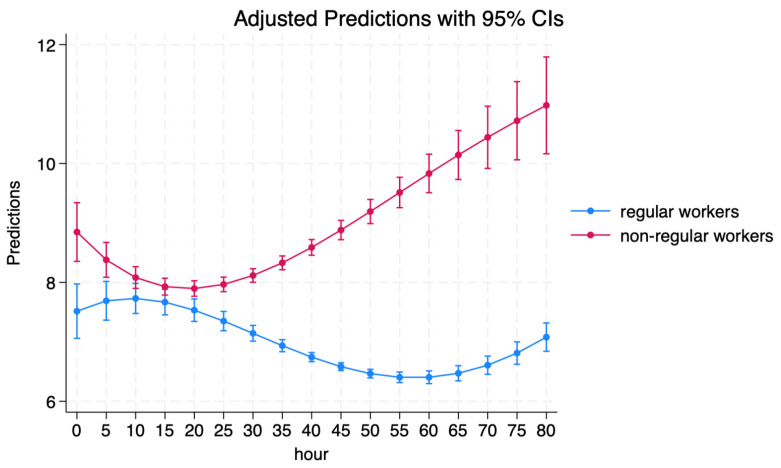
The raw relationship between work hours and MI. Source: Calculated based on data from the CSLC of 2010, 2013, 2016, and 2019 conducted by the MHLW. Notes: The calculations are based on the model as $M_{i}=\alpha +\beta_{1}{WH}_{i}+\beta_{2}WH_{i}^{2}+\beta_{3}{WH}_{i}^{3}+\beta_{4}{HW}_{i}^{4}+\epsilon_{i}$; *WH* represents the work hours. Non-regular workers include part-time, temporary, dispatched, contracted, and entrusted workers.

**Table 1 ijerph-21-00842-t001:** Descriptive statistics of variables.

	Weekly Working Hours ≥ 55 h (*N* = 5931)		Weekly Working Hours < 55 h (*N* = 52,446)		Gap
	Mean (a)	S.D.	Mean (b)	S.D.	a–b
Mental health score	3.536	4.387	3.204	4.086	0.332
Weekly work hours	64.375	7.066	36.264	12.785	28.111
Female dummy	0.121	0.327	0.489	0.500	0.368
Log of age	3.772	0.239	3.863	0.260	−0.091
Having a spouse	0.083	0.276	0.380	0.485	0.297
Number of children	0.018	0.136	0.017	0.136	0.001
Log of family income	6.432	0.568	6.325	0.662	0.107
Family Income	721.765	418.416	673.128	403.529	48.637
**Employment status**					
Non-regular worker	0.070	0.255	0.414	0.493	−0.344
Regular worker	0.930	0.255	0.586	0.493	0.344
**Occupation**					
Managers	0.114	0.318	0.065	0.247	0.049
Professional	0.312	0.463	0.259	0.438	0.053
Clerk	0.067	0.251	0.172	0.377	−0.105
Sale job	0.102	0.302	0.076	0.265	0.026
Service job	0.113	0.317	0.167	0.373	−0.054
Security job	0.030	0.170	0.016	0.125	0.014
Agriculture, forestry, and fishery job	0.010	0.100	0.010	0.097	0.000
Elementary job	0.229	0.420	0.198	0.399	0.031
Not elsewhere classified	0.022	0.148	0.037	0.188	−0.015
**Firm size**					
1–4	0.029	0.168	0.046	0.209	−0.017
5–29	0.185	0.388	0.203	0.402	−0.018
30–99	0.176	0.380	0.173	0.378	0.003
100–299	0.144	0.351	0.147	0.354	−0.003
200–499	0.063	0.244	0.063	0.242	0.000
500–999	0.062	0.242	0.069	0.253	−0.007
1000–4999	0.107	0.309	0.104	0.306	0.003
5000–	0.103	0.304	0.105	0.307	−0.002
Government office	0.131	0.337	0.091	0.288	0.040
**Spouse’s type of employment status**					
Not in work	0.494	0.500	0.458	0.498	0.036
Regular worker	0.213	0.410	0.342	0.474	−0.129
Part-time worker	0.224	0.417	0.133	0.339	0.091
Temporary worker	0.021	0.144	0.017	0.130	0.004
Dispatched worker	0.010	0.101	0.006	0.080	0.004
Contract worker	0.024	0.154	0.027	0.162	−0.003
Entrusted worker	0.009	0.095	0.012	0.110	−0.003
Other	0.004	0.062	0.004	0.066	0.000
**Scale of resident city (thousands)**					
Large city (more than 150)	0.262	0.440	0.228	0.419	0.034
Population Scale 150	0.311	0.463	0.298	0.457	0.013
Population Scale 50–149	0.255	0.436	0.272	0.445	−0.017
Population Scale 149 or less	0.066	0.249	0.085	0.279	−0.019
County	0.106	0.308	0.116	0.321	−0.010
**Survey year**					
2010	0.261	0.439	0.223	0.416	0.038
2013	0.297	0.457	0.274	0.446	0.023
2016	0.248	0.432	0.255	0.436	−0.007
2019	0.193	0.395	0.248	0.432	−0.055

Source: Calculated based on data from the CSLC of 2010, 2013, 2016, and 2019 conducted by the MHLW.

**Table 2 ijerph-21-00842-t002:** Average work hours by heterogeneous groups.

	Mean	SD	Min	Max
(1) Gender				
Male	45.05	13.18	0	89
Female	31.92	13.81	0	88
(2) Age				
Aged 16–29	40.52	16.25	0	89
Aged 30–49	41.22	15.17	0	89
Aged 50 and above	36.95	14.31	0	89
(3) Education				
Low education	38.09	14.41	0	89
Middle education	36.29	15.06	0	89
High education	43.24	15.03	0	89
(4) Occupation				
Non-managers	38.58	15.04	0	89
Managers	46.32	11.75	0	85
(5) Region				
Small cities	39.14	15.46	0	89
Large cities	39.09	14.40	0	88
(6) Employment status				
Non-Regular workers	28.43	12.91	0	88
Regular workers	45.81	11.92	0	89

Source: Calculated based on data from the CSLC of 2010, 2013, 2016, and 2019 conducted by the MHLW.

**Table 3 ijerph-21-00842-t003:** Results based on the OLS method.

	(1)		(2)		(3)		(4)		(5)	
Regular	0.249	***	−0.155	***	−0.187	***	−0.175	***	−0.158	***
	(3.04)		(−10.69)		(−14.45)		(−14.3)		(−13.07)	
WH	−0.038	***								
	(−7.33)									
WH*Regular	−0.047	***								
	(−10.45)									
WH^2^	0.001	***								
	(6.45)									
WH^2^*Regular	0.001	***								
	(6.26)									
WH^3^	−0.000	***								
	(−4.25)									
WH^2^*Regular	0.000	**								
	(2.07)									
Ref. 45WH										
40WH			0.218	***						
			(7.36)							
40WH*Regular			0.105	***						
			(3.20)							
50WH					0.386	***				
					(9.01)					
50WH*Regular					−0.125	***				
					(−2.76)					
55WH							0.472	***		
							(7.79)			
55WH*Regular							−0.086			
							(−1.36)			
60WH									0.617	***
									(8.30)	
60WH*Regular									−0.238	***
									(−3.08)	
Control variables	Yes		Yes		Yes		Yes		Yes	
Number of Observations	549,524		549,524		549,524		549,524		549,524	
Adjuster R-square	0.002		0.001		0.001		0.001		0.001	
Log Likelihood	−1,566,039		−1,566,407		−1,566,274		−1,566,217		−1,566,265	
F statistics	42.44		16.68		28.79		33.96		29.55	
Prob>F	0		0		0		0		0	

Source: Calculated based on data from the CSLC of 2010, 2013, 2016, and 2019 conducted by the MHLW. Note: *t*-values are in parentheses. ** *p* < 0.05, *** *p* < 0.01. *t*-values are in parentheses.

**Table 4 ijerph-21-00842-t004:** Probability of becoming workers with long work hours (regular worker).

	(1)		(2)		(3)		(4)	
	40WH		50WH		55WH		60WH	
Female	−0.462	***	−0.512	***	−0.559	***	−0.609	***
	(−17.65)		(−17.71)		(−15.80)		(−15.05)	
Ln age	−0.689	***	−0.811	***	−0.720	***	−0.737	***
	(−22.45)		(−25.94)		(−20.65)		(−19.71)	
Having a spouse	−0.190	***	−0.173	***	−0.133	***	−0.123	**
	(−6.17)		(−5.02)		(−3.13)		(−2.51)	
Ln family income	0.198	***	0.236	***	0.199	***	0.174	***
	(13.49)		(14.85)		(10.75)		(8.62)	
Occupation [Manager]								
Professional	−0.081	***	−0.089	***	−0.053	*	−0.056	*
	(−3.21)		(−3.56)		(−1.88)		(−1.84)	
Clerk	−0.381	***	−0.462	***	−0.456	***	−0.450	***
	(−13.36)		(−15.51)		(−12.95)		(−11.41)	
Sales workers	0.145	***	0.169	***	0.209	***	0.230	***
	(4.26)		(5.01)		(5.68)		(5.82)	
Service workers	−0.069	**	−0.053		0.029		0.085	**
	(−2.14)		(−1.64)		(0.79)		(2.16)	
Protective service workers	−0.258	***	−0.339	***	−0.165	***	−0.085	
	(−4.79)		(−6.20)		(−2.76)		(−1.34)	
Agriculture, forestry and fishery workers	0.025		−0.124		0.029		−0.002	
	(0.30)		(−1.44)		(0.30)		(−0.02)	
Elementary occupations	−0.039		−0.165	***	−0.060	**	−0.026	
	(−1.43)		(−6.05)		(−1.96)		(−0.78)	
Not elsewhere classified	−0.264	***	−0.230	***	−0.146	**	−0.104	
	(−5.04)		(−4.24)		(−2.38)		(−1.58)	
Firm size (number of employees)								
5–29	0.114	***	0.138	***	0.190	***	0.178	***
	(2.86)		(3.17)		(3.70)		(3.18)	
30–99	0.067	*	0.153	***	0.191	***	0.181	***
	(1.66)		(3.48)		(3.70)		(3.21)	
100–299	−0.033		0.098	**	0.092	*	0.043	
	(−0.82)		(2.21)		(1.75)		(0.76)	
200–499	−0.060		0.071		0.058		0.016	
	(−1.33)		(1.46)		(1.00)		(0.26)	
500–999	−0.067		0.098	**	0.025		−0.047	
	(−1.48)		(2.03)		(0.44)		(−0.76)	
1000–4999	−0.087	**	0.027		0.006		−0.025	
	(−2.06)		(0.58)		(0.10)		(−0.43)	
5000-	−0.130	***	−0.024		−0.056		−0.133	**
	(−3.09)		(−0.53)		(−1.05)		(−2.26)	
Government office	0.024		0.240	***	0.336	***	0.300	***
	(0.56)		(5.15)		(6.20)		(5.05)	
Spouse’s employment status [non-work]								
Regular worker	−0.102	***	−0.115	***	−0.128	***	−0.111	***
	(−5.25)		(−5.69)		(−5.57)		(−4.39)	
Part-time	0.054	***	0.007		−0.024		0.017	
	(2.79)		(0.37)		(−1.10)		(0.72)	
Temporary worker	0.022		−0.006		0.012		0.061	
	(0.42)		(−0.12)		(0.20)		(0.98)	
Dispatched worker	0.048		0.072		0.022		0.118	
	(0.60)		(0.91)		(0.25)		(1.29)	
Contract worker	0.019		0.060		0.016		−0.019	
	(0.40)		(1.27)		(0.30)		(−0.32)	
Entrusted worker	0.015		0.009		0.021		0.018	
	(0.20)		(0.13)		(0.26)		(0.20)	
Other	−0.004		0.094		0.070		0.050	
	(−0.03)		(0.84)		(0.57)		(0.36)	
Number of children	0.027		−0.015		0.057		0.102	*
	(0.53)		(−0.29)		(0.99)		(1.68)	
City Scale [Large city] (thousands)								
Population 150	−0.047	**	−0.070	***	−0.046	**	−0.073	***
	(−2.50)		(−3.63)		(−2.15)		(−3.15)	
Population 50−149	−0.080	***	−0.111	***	−0.103	***	−0.118	***
	(4.12)		(−5.55)		(−4.55)		(−4.84)	
Population 149 or less	−0.101	***	−0.169	***	−0.163	***	−0.234	***
	(−3.57)		(−5.75)		(−4.82)		(−6.21)	
County	−0.097	***	−0.148	***	−0.126	***	−0.173	***
	(−3.89)		(−5.74)		(−4.30)		(−5.38)	
Constant term	2.480	***	2.153	***	1.435	***	1.514	***
	(17.94)		(15.33)		(9.03)		(8.76)	
Number of observations	35,870		35,870		35,870		35,870	
Pseudo-R-Square	0.063		0.071		0.067		0.073	
Log Likelihood	−23,109		−21,585		−16,013		−13,163	
Chi2 Statistics	2868.900		2830.200		1907.500		1613.600	
Prob > Chi2	0.000		0.000		0.000		0.000	

Source: Calculated based on data from the CSLC of 2010, 2013, 2016, and 2019 conducted by the MHLW. Note: *t*-values are in parentheses. * *p* < 0.1, ** *p* < 0.05, *** *p* < 0.01. *t*-values are in parentheses.

**Table 5 ijerph-21-00842-t005:** Probability of become workers with long work hours (non-regular worker).

	(1)		(2)		(3)		(4)	
	40WH		50WH		55WH		60WH	
Female	−0.200	***	−0.317	***	−0.307	***	−0.432	***
	(−4.68)		(−5.65)		(−4.15)		(−4.91)	
Ln age	−0.699	***	−0.762	***	−0.690	***	−0.707	***
	(−13.79)		(−12.17)		(−8.80)		(−7.80)	
Spouse of household	−0.357	***	−0.207	***	−0.160	*	−0.149	
	(−7.40)		(−3.21)		(−1.86)		(−1.41)	
Ln family income	0.102	***	0.115	***	0.125	***	0.106	**
	(4.69)		(4.15)		(3.55)		(2.48)	
**Type of work [Part-time]**								
Dispatched worker	0.010		−0.020		0.050		−0.070	
	(0.21)		(−0.37)		(0.68)		(−0.76)	
Contract worker	0.450	***	0.380	***	0.350	***	0.330	***
	(8.16)		(5.60)		(3.83)		(3.08)	
Entrusted Worker	0.660	***	0.520	***	0.460	***	0.390	***
	(19.65)		(12.18)		(8.45)		(5.95)	
Occupation [Manager]								
Professional	0.069		0.130		0.032		0.077	
	(0.76)		(1.17)		(0.23)		(0.44)	
Clerk	−0.060		−0.073		−0.218		−0.306	
	(−0.64)		(−0.62)		(−1.44)		(−1.53)	
Sale	0.038		0.048		0.080		0.162	
	(0.39)		(0.39)		(0.52)		(0.84)	
Service	0.101		0.136		0.195		0.310	*
	(1.11)		(1.20)		(1.38)		(1.75)	
Security	0.375	***	0.417	***	0.360	**	0.435	**
	(3.04)		(2.85)		(1.97)		(2.03)	
Agriculture, forestry and Fishery workers	0.341	**	0.394	**	0.285		−0.132	
	(2.52)		(2.38)		(1.37)		(−0.42)	
Elementary	0.356	***	0.272	**	0.210		0.242	
	(3.97)		(2.45)		(1.51)		(1.40)	
Not elsewhere classified	0.036		0.022		0.019		0.135	
	(0.35)		(0.17)		(0.12)		(0.68)	
Firm size [1–4] (number of employees)								
5–29	−0.075		−0.123		−0.145		−0.053	
	(−1.26)		(−1.61)		(−1.52)		(−0.45)	
30–99	−0.017		−0.036		−0.063		−0.013	
	(−0.27)		(−0.48)		(−0.66)		(−0.11)	
100–299	−0.018		−0.058		−0.083		−0.080	
	(−0.29)		(−0.73)		(−0.84)		(−0.65)	
200–499	−0.033		−0.096		−0.177		−0.128	
	(−0.44)		(−1.02)		(−1.48)		(−0.89)	
500–999	−0.025		−0.074		−0.231	**	−0.218	
	(−0.35)		(−0.81)		(−1.97)		(−1.52)	
1000–4999	−0.072		−0.146	*	−0.255	**	−0.292	**
	(−1.07)		(−1.67)		(−2.28)		(−2.09)	
5000–	−0.222	***	−0.212	**	−0.317	***	−0.348	**
	(−3.11)		(−2.35)		(−2.70)		(−2.30)	
Government office	−0.560	***	−0.347	***	−0.224	*	−0.230	
	(−6.57)		(−3.35)		(−1.75)		(−1.44)	
Spouse’s employment status [non-work]								
Regular worker	−0.226	***	−0.284	***	−0.297	***	−0.282	***
	(−5.63)		(−5.43)		(−4.32)		(−3.34)	
Part-time	0.058		0.076		0.071		0.042	
	(1.32)		(1.44)		(1.08)		(0.55)	
Temporary worker	0.060		−0.038		0.044		0.184	
	(0.68)		(−0.33)		(0.32)		(1.26)	
Dispatched worker	0.060		0.224		0.008		−0.397	
	(0.38)		(1.30)		(0.04)		(−1.04)	
Contract worker	−0.012		−0.041		0.024		−0.071	
	(−0.17)		(−0.45)		(0.22)		(−0.51)	
Entrusted worker	−0.106		−0.060		−0.067		−0.099	
	(−0.92)		(−0.43)		(−0.37)		(−0.42)	
Other	−0.243		−0.007					
	(−1.07)		(−0.03)					
Number of children	0.088		0.202	*	0.204		0.193	
	(0.99)		(1.85)		(1.52)		(1.16)	
City Scale [Large city] (thousands)								
Population 150	0.060	*	0.006		0.049		0.007	
	(1.73)		(0.15)		(0.87)		(0.10)	
Population 50–149	0.083	**	−0.020		−0.037		−0.054	
	(2.35)		(0.43)		(−0.62)		(−0.77)	
Population 149 or less	0.128	**	0.083		0.078		−0.051	
	(2.57)		(1.32)		(0.95)		(0.49)	
County	0.196	***	0.097	*	0.161	**	0.086	
	(4.46)		(1.73)		(2.30)		(1.04)	
Constant term	1.422	***	1.249	***	0.540		0.677	
	(5.93)		(4.22)		(1.48)		(1.58)	
Number of observations	21,884		21,884		21,798		21,798	
Pseudo-R-Square	0.135		0.122		0.110		0.124	
Log Likelihood	−6278		−3577		−2030		−1353	
Chi2 Statistics	1803.500		848.600		423.300		303.100	
Prob>Chi2	0.000		0.000		0.000		0.000	

Source: Calculated based on data from the CSLC of 2010, 2013, 2016, and 2019 conducted by the MHLW. Note: *t*-values are in parentheses. * *p* < 0.1, ** *p* < 0.05, *** *p* < 0.01. *t*-values are in parentheses.

**Table 6 ijerph-21-00842-t006:** Results based on the PSM method.

	(1)		(2)	
	Coef.	SE.	Coef.	SE.
(1) 50WH				
ATT	0.306	0.048	0.403	0.051
Covariates	No		Yes	
(2) 55WH				
ATT	0.353	0.059	0.454	0.061
Covariates	No		Yes	
(3) 60WH				
ATT	0.324	0.06	0.435	0.069
Covariates	No		Yes	

Source: Calculated based on data from the CSLC of 2010, 2013, 2016, and 2019 conducted by the MHLW. Note: PSM method was used. We used 45 h weekly as the control group. Covariate variables included gender, age, income, occupation, employment status, firm size, and city size.

**Table 7 ijerph-21-00842-t007:** Results by regular and non-regular workers based on the PSM method.

	(1) Regular			(2) No-Regular		
	Coef.		SE.	Coef.		SE.
ATT	0.457	***	0.061	0.518	**	0.277
Covariates	Yes			Yes		

Source: Calculated based on data from the Comprehensive Survey of Living Conditions (CSLC) of 2010, 2013, 2016, and 2019 conducted by the MHLW. ** *p* < 0.05, *** *p* < 0.01.

**Table 8 ijerph-21-00842-t008:** Results by gender and work-related groups based on the PSM method.

**(a) gender**									
	**Men**			**(2) Women**					
	Coef.		SE.	Coef.		SE.			
ATT	0.43	***	0.063	0.718	***	0.16			
Covariates	Yes			Yes					
**(b) occupation**									
	**(1) Manager**			**(2) Non-Manager**					
	Coef.		SE.	Coef.		SE.			
ATT	0.767	***	0.161	0.43	***	0.063			
Covariates	Yes			Yes					
**(c) firm size**									
	**(1) Small (1-99)**			**(2) Middle (100–299)**			**(3) Large (300 or more)**		
	Coef.		SE.	Coef.		SE.	Coef.		SE.
ATT	0.547	***	0.135	0.285	***	0.105	0.612	***	0.098
Covariates	Yes			Yes			Yes		
**(d) city size**									
	**(1) Large city**			**(2) Middle-size city**			**(3) Small-size city**		
	Coef.		SE.	Coef.		SE.	Coef.		SE.
ATT	0.415	***	0.078	0.531	***	0.089	0.547	***	0.135
Covariates	Yes			Yes			Yes		

Source: Calculated based on data from the CSLC of 2010, 2013,2016, and 2019 conducted by the MHLW. Note: PSM method was used. We used 55 h or more as an indicator of long working hours in the analysis. The covariate variables included sex, age, income, occupation, employment status, firm size, and city size. *** *p* < 0.01.

## Data Availability

The dataset used in this study, the Comprehensive Survey of Living Conditions (CSLC) of 2010, 2013, 2016, and 2019, was managed by the Ministry of Health, Labour and Welfare (MHLW) of Japan. https://www.mhlw.go.jp/toukei/list/20-21.html.

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
