# Peer review of "Impact of Long Working Hours on Mental Health Status in Japan: Evidence from a National Representative Survey"

_ijerph, 2024, doi:10.3390/ijerph21070842_

Round 1

Reviewer 1 Report

Comments and Suggestions for Authors

The topic addressed in the article, concerning the impact of long working hours on mental health in Japan, is relevant and timely, especially in the context of the country's specific organisational culture. The study, based on a nationally representative survey conducted between 2003 and 2019, provides valuable information on this phenomenon. The authors present three main findings, which are clear and have important practical implications for government policy and personnel management.

However, the reviewer notes several important issues that may require attention and improvement:

The literature review: I therefore propose to take up the suggestion that the literature review needs to be redesigned in order to develop the research hypotheses. The focus should be on a more precise articulation of the existing theories and research that underpin the hypotheses.

Explanation of abbreviations: It is important that abbreviations used for the first time, such as the OLS method, should be explained to make it easier to understand for readers, especially those who are not statistical specialists. Please go through the whole text with this in mind.

Discussion section: The absence of a discussion section in the article is a significant shortcoming. This section is essential in order to verify the hypotheses set out, to discuss the research results obtained in the context of the literature and to interpret the results in terms of their practical implications. The suggestion to add this section is even welcome.

In conclusion, the article brings important findings on the impact of long working hours on mental health in Japan, but needs some improvements, especially in terms of literature review, clarification of abbreviations and the addition of a discussion section. Improving these elements could significantly enhance the value and comprehensibility of the article.

Author Response

Response to Comments

Reviewer1

Thank you very much for your thoughtful suggestions and insights. The manuscript has greatly benefited from these valuable comments.

We have carefully reviewed all the comments and made the necessary changes accordingly. Please find the responses to each comment attached below. The revised parts were expressed by using red color.

  1. The literature review: I therefore propose to take up the suggestion that the literature review needs to be redesigned in order to develop the research hypotheses. The focus should be on a more precise articulation of the existing theories and research that underpin the hypotheses.

[Reply] Thank you very much for your helpful comments. Following your suggestions, we completely corrected the contents of Introduction part to clearly describe the importance and our motivations of this study as follows (pp.1-2):

  1. Introduction

Mental illness is often referred to as a mental disorder, and long working hours are two significant issues affecting employees worldwide [1–3]. In 2017, approximately 792 million individuals worldwide were reported to be living with mental illness, accounting for 10.7% of the global population, slightly exceeding 1 in 10 people [2]. Given the high medical care expenses associated with mental illness [2–3], and considering that mental illness can reduce labor productivity, thereby negatively affecting human capital accumulation in most countries, exploring the determinants of mental illness has become a critical issue in the field of public health.

Several empirical studies have shed light on potential determinants of mental illness, including individual attributes such as education, age, and sex [4–7]. Additionally, social capital has been identified as a factor that influences the risk of developing mental illness [8–9] along with life events such as marriage and fertility [10–12]. Understanding these factors can contribute significantly to addressing mental illness challenges and implementing effective public health interventions.

Furthermore, work-life conflict has been identified as a factor that may increase the risk of mental illness [13, 14]. Empirical research has consistently highlighted long working hours as a significant risk factor among the various work environment factors that can adversely affect workers’ mental health. Numerous studies have found a strong association between long working hours and a higher risk of developing mental disorders in both developing and developed countries [4–7, 15–18]. However, the evidence from Japan is scarce.

In Japan, the prevalence of workplace stress among workers is steadily increasing, which has implications for mental health. According to the Survey on the State of Employees' Health conducted by the Ministry of Health, Labour and Welfare (MHLW), the percentage of workers experiencing strong anxiety, worry, or stress in their work or working life increased from 55.0% in 1987 to 61.5% in 2002. This rise in workplace stress is also evident in the alarming increase in the number of patients with mental disorders and suicides nationwide. The total number of patients with mental disorders in Japan increased from 4.33 million in 2013 to 17.21 million in 2022. The total number of suicides in Japan surged from 27, 282 in 2013 to 21,881 in 2022, even as the number of employed persons will decrease from 9,401 in 2013 to 7,862 in 2016 and increase to 8,576 in 2022. The 2008 White Paper on Suicide Countermeasures published by the Cabinet Office revealed that "health problems" accounted for the largest proportion of causes and motives for suicide (63.3 %). Long working hours in Japan are a critical factor contributing to these issues. In 2004, the MHLW published a report on the “Study Group on Overwork and Mental Health Countermeasures,” which selected employees based on the fact that they had worked long overtime hours. It recommends creating a mechanism to check employee health. Recently, the Japanese government promoted the reduction of long working hours and improved mental health in the workplace. However, when compared internationally, Japan, along with the United States, had the longest average annual working hours of 1,800 hours or more among developed countries during the 2000s. Therefore, investigating the association between long working hours and mental illnesses, particularly in Japan, is of the utmost importance.

This study focused on this issue in Japan. Some studies have investigated this issue from the perspectives of occupational health and epidemiology. Fujino et al. [19] conducted a systematic literature review of 17 papers that addressed this issue. Recent survey data have been utilized by other researchers [20–24], who have also examined the relationship between long working hours and the likelihood of developing mental disorders, such as depression and stress, in Japan. However, many of these studies, from occupational health or medical perspectives, have not adequately considered endogeneity. However, few empirical studies have explored this issue from an economic perspective. Three of these studies are particularly relevant to this study. Two studies investigated the association between long working hours and mental health, using longitudinal survey data and employing fixed effect (FE) models to address individual heterogeneity [25–26]. Okamoto used longitudinal survey data and instrumental variable (IV) methods to discover that long working hours may lead to reduced sleep hours and an increased probability of becoming obese [27]. However, these three studies did not adequately account for non-random sample selection issues between the long and non-long working-hour groups. This issue is related to endogeneity and may introduce a measurement bias into the empirical results. Additionally, although there are large disparities in work conditions (e.g., earned wages, work hours, and social insurance) in Japan between regular and non-regular workers (e.g., part-time, temporary, dispatched, contract, and entrusted workers) [28–30] and it has been noted that the working hours of non-regular workers are also long in Japan [31], previous studies have not explored the differences in the effects of long working hours on mental illness between regular and non-regular workers. This study addresses these gaps in the literature.

  1. Explanation of abbreviations: It is important that abbreviations used for the first time, such as the OLS method, should be explained to make it easier to understand for readers, especially those who are not statistical specialists. Please go through the whole text with this in mind.

[Reply] Thank you very much for your helpful comments. We added the contents to introduce the OLS method as follows (p.4):

As a benchmark, we employ the ordinary least squares (OLS) method to estimate the association between long working hours and mental illness, as expressed in Eq. (1). The ordinary least squares (OLS) method is a type of linear least squares method used to estimate unknown parameters in a linear regression model by minimizing the sum of the squares of the differences between the observed dependent variable in the dataset and the predicted values from the linear function of the independent variables.

  1. Discussion section: The absence of a discussion section in the article is a significant shortcoming. This section is essential in order to verify the hypotheses set out, to discuss the research results obtained in the context of the literature and to interpret the results in terms of their practical implications. The suggestion to add this section is even welcome.

[Reply] Thank you very much for your helpful comments. Following your suggestions, we added the new section (6. Discussions) to discuss the research results and policy implications as follows (pp.15-16):

  1. 6. Discussions

6.1 Novel Findings of This Study

This study investigated the effects of long working hours on mental health status in the Japanese context, considering a non-random sample selection bias among groups with different working hours. The results indicated that long working hours, especially weekly work hours exceeding 55 hours, significantly increased the likelihood of developing mental health disorders. These findings align with previous studies in Japan from occupational health and epidemiological perspectives [19–24] as well as from an economic perspective [25–27], highlighting the persistent negative impact of long working hours on individuals' well-being. Furthermore, the results are consistent with conclusions drawn from research conducted in other countries [4–7, 15–18].

We also compared the effects of long working hours on the mental health status of regular and non-regular workers, a comparison not previously considered in the literature. These new findings suggest that the negative effect of long working hours is more pronounced among non-regular workers than among regular workers. These findings can be explained by the effort-reward imbalance model [33], which suggests that when the efforts involved in involuntary long working hours are not adequately rewarded, the likelihood of experiencing mental illness may be higher among those working long hours. In Japan, there exists a wage gap between regular and non-regular workers [28–30]. When nonregular workers receive lower wages despite working similar hours to regular workers, the probability of experiencing mental illness among nonregular workers is likely to be higher than that among regular workers.

Additionally, the results regarding the differences in the effects of long working hours among gender, occupation, firm size, and city size groups demonstrated that the negative effects on mental health status were greater for women, managers, workers in small or large firms, and workers in small cities. We consider the reasons as follows: First, according to family economics theory, there exists a gender role division in households known as "the man as the breadwinner and the woman as the homemaker" [34–35]. As women often bear more responsibility for family care (e.g., childcare and elder care), the negative effect of long working hours on mental health status may be greater for women than for men. Second, in many Japanese firms, managers often do not receive overtime allowances, whereas non-managerial workers do. According to the effort-reward imbalance model [33], this disparity may lead to a greater negative effect of long working hours on managers than on non-managers. Third, there is a wage gap based on firm size in Japan [40–41], with workers in small firms typically earning lower wages than those in medium-sized firms. According to the effort-reward imbalance model, this wage disparity may result in a greater negative effect of long working hours for workers in small firms than in medium-sized firms. Although wages are higher in large firms than in medium firms, workers in large firms often face higher work responsibilities and lower work authority. According to the job demand-control model [32], this imbalance may lead to a greater negative effect of long working hours on workers’ mental health status in large firms than in medium firms. Finally, fierce business competition is more prevalent in large cities. Due to the intense competitive environment, the likelihood of achieving work-life balance may be lower for workers in large cities. Additionally, workers who prefer to avoid fierce competition may choose to work in smaller cities. Therefore, the negative effects of long working hours on mental health may be greater for workers in small cities than those in large cities.

These new empirical findings enrich our understanding of the related personnel and labor economic theories on this issue. This study also underscores the importance of distinguishing between different types of employment status (regular and non-regular workers) and heterogeneous groups (e.g., gender, occupation, firm size, and city size) for future research.

6.2 Practical Implications

Reviewer 2 Report

Comments and Suggestions for Authors

It was a joy reading this manuscript, which was well-written and shed light on a very pressing issue of organizational life in modern times. I also appreciated the authors' grand undertaking of the large size of data to tell the rich nuances of the story. In the following, I would like to share several observations, and hope you will find them helpful.

1. Research motive: to better present the rationale and contributions of the paper for the readers, it might be helpful to articulate them from the perspectives of phenomenon instead of methodology. The methodology is ultimately in service of the research question, not the other way around. In the abstract, introduction, literature review, you repeatedly indicated that your study aimed to address the "endogeneity issue" in current research. But never elaborated why this issue needed to be addressed. Put it another way, Why should readers care about this methodology issue? How has this endogeneity issue prevented us from understanding the phenomenon? What new insights would your approach provide readers about the phenomenon? 

2. Hypothesis development and testing: it was not clear to me that if you were trying to conduct a hypothesis testing study, or an exploratory study. 

If this was a hypothesis testing: then, the logic is to present what we know and what we don't know about a phenomenon. Then, using theories to predict a relationship (e.g., long hours related to stress), which is followed by data analysis to test this hypothesis. 

This was not really the case in the writing. You actually said from very beginning that studies showed long hours predicted stress, and the only issue was the methodology issue. You did not provide any hypothesis.

The way you defined "long hours" (55 hours) was also exploratory considering you drew a conclusion based on the analysis result, without any conceptualizing.  

If this were an exploratory study, then the writing and framing must align with this approach. 

You should articulate why those factors contribute to the variations in the long-hour effects. For a hypothesis testing study, this explanation should be part of the hypothesis development and come before your data analysis. For an exploratory study, this explanation should come after the result. 

Finally, the literature review section was mostly repeating what you wrote in the introduction section instead of hypothesis development. Again, the goal and structure of the writing for this part would vary depending on weather you are conducting a hypothesis testing or an exploratory study.

3. Data analysis transparency: you mentioned that the data was from 2003 - 2019. What does this mean? How did you treat the data for each year? 

Comments on the Quality of English Language

You are very good writers - the paper was written beautifully. There were only a few places that could use editing. 

Author Response

Response to Comments

Reviewer 2#

Thank you very much for your thoughtful suggestions and insights. The manuscript has greatly benefited from these valuable comments.

We have carefully reviewed all the comments and made the necessary changes accordingly. Please find the responses to each comment attached below. The revised parts were expressed by using red color.

  1. Research motive: to better present the rationale and contributions of the paper for the readers, it might be helpful to articulate them from the perspectives of phenomenon instead of methodology. The methodology is ultimately in service of the research question, not the other way around. In the abstract, introduction, literature review, you repeatedly indicated that your study aimed to address the "endogeneity issue" in current research. But never elaborated why this issue needed to be addressed. Put it another way, Why should readers care about this methodology issue? How has this endogeneity issue prevented us from understanding the phenomenon? What new insights would your approach provide readers about the phenomenon? 

[Reply] Thank you very much for your helpful comments. Following your suggestions, we completely corrected the contents of Introduction part to clearly describe the importance and our motivations of this study as follows (pp.1-2):

  1. Introduction

Mental illness is often referred to as a mental disorder, and long working hours are two significant issues affecting employees worldwide [1–3]. In 2017, approximately 792 million individuals worldwide were reported to be living with mental illness, accounting for 10.7% of the global population, slightly exceeding 1 in 10 people [2]. Given the high medical care expenses associated with mental illness [2–3], and considering that mental illness can reduce labor productivity, thereby negatively affecting human capital accumulation in most countries, exploring the determinants of mental illness has become a critical issue in the field of public health.

Several empirical studies have shed light on potential determinants of mental illness, including individual attributes such as education, age, and sex [4–7]. Additionally, social capital has been identified as a factor that influences the risk of developing mental illness [8–9] along with life events such as marriage and fertility [10–12]. Understanding these factors can contribute significantly to addressing mental illness challenges and implementing effective public health interventions.

Furthermore, work-life conflict has been identified as a factor that may increase the risk of mental illness [13, 14]. Empirical research has consistently highlighted long working hours as a significant risk factor among the various work environment factors that can adversely affect workers’ mental health. Numerous studies have found a strong association between long working hours and a higher risk of developing mental disorders in both developing and developed countries [4–7, 15–18]. However, the evidence from Japan is scarce.

In Japan, the prevalence of workplace stress among workers is steadily increasing, which has implications for mental health. According to the Survey on the State of Employees' Health conducted by the Ministry of Health, Labour and Welfare (MHLW), the percentage of workers experiencing strong anxiety, worry, or stress in their work or working life increased from 55.0% in 1987 to 61.5% in 2002. This rise in workplace stress is also evident in the alarming increase in the number of patients with mental disorders and suicides nationwide. The total number of patients with mental disorders in Japan increased from 4.33 million in 2013 to 17.21 million in 2022. The total number of suicides in Japan surged from 27, 282 in 2013 to 21,881 in 2022, even as the number of employed persons will decrease from 9,401 in 2013 to 7,862 in 2016 and increase to 8,576 in 2022. The 2008 White Paper on Suicide Countermeasures published by the Cabinet Office revealed that "health problems" accounted for the largest proportion of causes and motives for suicide (63.3 %). Long working hours in Japan are a critical factor contributing to these issues. In 2004, the MHLW published a report on the “Study Group on Overwork and Mental Health Countermeasures,” which selected employees based on the fact that they had worked long overtime hours. It recommends creating a mechanism to check employee health. Recently, the Japanese government promoted the reduction of long working hours and improved mental health in the workplace. However, when compared internationally, Japan, along with the United States, had the longest average annual working hours of 1,800 hours or more among developed countries during the 2000s. Therefore, investigating the association between long working hours and mental illnesses, particularly in Japan, is of the utmost importance.

This study focused on this issue in Japan. Some studies have investigated this issue from the perspectives of occupational health and epidemiology. Fujino et al. [19] conducted a systematic literature review of 17 papers that addressed this issue. Recent survey data have been utilized by other researchers [20–24], who have also examined the relationship between long working hours and the likelihood of developing mental disorders, such as depression and stress, in Japan. However, many of these studies, from occupational health or medical perspectives, have not adequately considered endogeneity. However, few empirical studies have explored this issue from an economic perspective. Three of these studies are particularly relevant to this study. Two studies investigated the association between long working hours and mental health, using longitudinal survey data and employing fixed effect (FE) models to address individual heterogeneity [25–26]. Okamoto used longitudinal survey data and instrumental variable (IV) methods to discover that long working hours may lead to reduced sleep hours and an increased probability of becoming obese [27]. However, these three studies did not adequately account for non-random sample selection issues between the long and non-long working-hour groups. This issue is related to endogeneity and may introduce a measurement bias into the empirical results. Additionally, although there are large disparities in work conditions (e.g., earned wages, work hours, and social insurance) in Japan between regular and non-regular workers (e.g., part-time, temporary, dispatched, contract, and entrusted workers) [28–30] and it has been noted that the working hours of non-regular workers are also long in Japan [31], previous studies have not explored the differences in the effects of long working hours on mental illness between regular and non-regular workers. This study addresses these gaps in the literature.

  1. Hypothesis development and testing: it was not clear to me that if you were trying to conduct a hypothesis testing study, or an exploratory study. 

If this was a hypothesis testing: then, the logic is to present what we know and what we don't know about a phenomenon. Then, using theories to predict a relationship (e.g., long hours related to stress), which is followed by data analysis to test this hypothesis. 

This was not really the case in the writing. You actually said from very beginning that studies showed long hours predicted stress, and the only issue was the methodology issue. You did not provide any hypothesis.

The way you defined "long hours" (55 hours) was also exploratory considering you drew a conclusion based on the analysis result, without any conceptualizing.  

If this were an exploratory study, then the writing and framing must align with this approach. 

You should articulate why those factors contribute to the variations in the long-hour effects. For a hypothesis testing study, this explanation should be part of the hypothesis development and come before your data analysis. For an exploratory study, this explanation should come after the result. 

Finally, the literature review section was mostly repeating what you wrote in the introduction section instead of hypothesis development. Again, the goal and structure of the writing for this part would vary depending on weather you are conducting a hypothesis testing or an exploratory study.

[Reply] Thank you very much for your helpful comments. According to your suggestions, we completely corrected the contents of Literature Review part to clearly describe the hypotheses development in this study as follows (pp.3-4):

  1. Literature Review and Hypotheses Development

Regarding the relationship between long working hours and mental illness, two models from the occupational health and personal economics perspectives have been proposed to suggest that long working hours negatively affect mental health status. First, the job demand-control model [32] posits that involuntary long working hours may lead to mental illness because of an imbalance between work responsibility (the reality of long working hours) and authority (wherein employees lack the authority to determine their working hours). Second, according to the effort-reward imbalance model [33], when the efforts involved in involuntary long working hours are not adequately rewarded (e.g., through unpaid overtime or a low overtime premium), the probability of developing mental illness may be higher among those working long hours.

From the perspective of labor and family economics theory based on the individual/household utility model, individuals aim to maximize their utility while considering income and time constraints. An individual's time allocation can be divided into three parts: market work, housework, and leisure [34–35]. This division implies a trade-off between working hours and hours devoted to housework and leisure activities. Consequently, long working hours may reduce the time available for housework and leisure, leading to work-life conflicts. This conflict may be particularly pronounced for female workers during their motherhood, as they face additional responsibilities and time constraints. This reduction in housework/leisure time and the resulting work-life conflict may ultimately lead to a decline in an individual's utility and increase the risk of developing mental illness [36–37].

Numerous studies have found that long working hours are a primary contributor to mental illness among working-age populations, in addition to demographic factors [4–7, 15–18], family-related factors [10–12], and social factors such as social participation and social capital [8–9].

Some studies have reported that long working hours negatively affect mental health [4–7, 15–18]. In Japan, there are a series of studies focusing on this issue from occupational health, epidemiology [19–24], and economic perspectives [2527]. Their findings indicated that long working hours may increase the likelihood of developing mental disorders, such as depression and stress, in Japan. Few empirical studies have addressed this issue from an economic perspective, three of which are closely related to this study. Ma [25] utilized survey data from the 2004-2008 Keio Household Panel Survey (KHPS 2004-2008) and employed a fixed-effects model with lagged variables to address endogeneity concerns. The empirical analysis investigated the impact of long working hours on the mental health status of workers in Japan and found that long working hours increased the probability of developing mental disorders. Furthermore, the study found that the impact of long working hours on mental health status varies across different groups; it is more significant for workers in the private sector, workers in large firms, low-education workers, low-income workers, female workers, and male non-managerial occupational workers than for their counterparts. Kuroda and Yamamoto [26] used four-wave longitudinal data from the Survey of Companies and Employees on Human Capital Development and Work-Life Balance, which has been conducted annually since 2012 by the Research Institute of Economy, Trade, and Industry. Their investigation focused on how the number of work hours, job characteristics, and workplace circumstances affected workers' mental health. The study found that long working hours significantly contributed to deterioration in respondents' mental health status, even after controlling for individual fixed effects and other characteristics. Furthermore, the relationship between work hours and mental health status was not linear, as working more than 50 hours per week notably erodes workers’ mental health status. Using longitudinal survey data from the Japan Household Panel Survey/Keio Household Panel Survey of 2004-2017 and the instrumental variable (IV) method, Okamoto [27] analyzed the effects of work hours on body mass index, smoking, and sleeping hours in Japan. This study found that longer working hours led to reduced sleep duration and increased the probability of obesity among workers.

According to personnel and labor economics theories and the empirical studies above, Hypothesis 1 (H1) is formulated as follows: 

Hypothesis 1: Individuals working longer hours exhibit a higher likelihood of developing mental illness compared to those working regular or fewer hours.

Several studies have demonstrated the persistence of the wage gap between regular and nonregular workers in Japan [28–30]. Some studies find that the wage gap between the two groups is due to disparities in endowment factors (e.g., human capital) and differences in the return to endowment (e.g., wage return on education), which are related to discrimination against nonregular workers in the workplace [28, 30]. According to the effort-reward imbalance model [33], when nonregular workers work similar hours but receive lower wages than their counterparts (regular workers), the negative effect of long working hours on mental health may be more pronounced among nonregular workers. Consequently, Hypothesis 2 (H2) was assumed to be

Hypothesis 2: The negative effect of long working hours on mental health status is more pronounced among non-regular workers than among regular worker groups.

In the following section, we conduct an empirical analysis to clarify these two hypotheses.

  1. Data analysis transparency: you mentioned that the data was from 2003 - 2019. What does this mean? How did you treat the data for each year? 

[Reply] Thank you very much for your helpful comments. Following your suggestions, we corrected the contents to introduce the data as follows (p.5):

3.2. Date and Variable Setting

This study used repeated cross-sectional data from the Comprehensive Survey of Living Conditions (CSLC), a nationally representative survey conducted by MHLW in Japan. The CSLC surveys households and their members throughout the country. We used the latest four waves of the CSLC, from 2010, 2013, 2016, and 2019, which provide rich information on mental health status, individual and household characteristics, and work status. Because the questions in the household questionnaire were nearly identical across these four waves, we pooled the data from the CSLC for these years in our analysis.

Round 2

Reviewer 2 Report

Comments and Suggestions for Authors

I appreciate authors' significant effort in revising the manuscript and responding to my comments with such diligence. The issues I raised have been addressed fully.